# QMDP-Net: Deep Learning for Planning under Partial Observability

**Peter Karkus**[1,2]     **David Hsu**[1,2]     **Wee Sun Lee**[2]

[1]NUS Graduate School for Integrative Sciences and Engineering
[2]School of Computing

National University of Singapore
{karkus, dyhsu, leews}@comp.nus.edu.sg

## Abstract

*This paper introduces the* QMDP-net, *a neural network architecture for planning under partial observability. The QMDP-net combines the strengths of model-free learning and model-based planning. It is a recurrent policy network, but it represents a policy for a parameterized set of tasks by connecting a model with a planning algorithm that solves the model, thus embedding the solution structure of planning in a network learning architecture. The QMDP-net is fully differentiable and allows for end-to-end training. We train a QMDP-net on different tasks so that it can generalize to new ones in the parameterized task set and "transfer" to other similar tasks beyond the set. In preliminary experiments, QMDP-net showed strong performance on several robotic tasks in simulation. Interestingly, while QMDP-net encodes the QMDP algorithm, it sometimes outperforms the QMDP algorithm in the experiments, as a result of end-to-end learning.*

## 1   Introduction

Decision-making under uncertainty is of fundamental importance, but it is computationally hard, especially under partial observability [24]. In a partially observable world, the agent cannot determine the state exactly based on the current observation; to plan optimal actions, it must integrate information over the past history of actions and observations. See Fig. 1 for an example. In the model-based approach, we may formulate the problem as a *partially observable Markov decision process* (POMDP). Solving POMDPs exactly is computationally intractable in the worst case [24]. Approximate POMDP algorithms have made dramatic progress on solving large-scale POMDPs [17, 25, 29, 32, 37]; however, manually constructing POMDP models or learning them from data remains difficult. In the model-free approach, we directly search for an optimal solution within a policy class. If we do not restrict the policy class, the difficulty is data and computational efficiency. We may choose a parameterized policy class. The effectiveness of policy search is then constrained by this a priori choice.

Deep neural networks have brought unprecedented success in many domains [16, 21, 30] and provide a distinct new approach to decision-making under uncertainty. The deep Q-network (DQN), which consists of a convolutional neural network (CNN) together with a fully connected layer, has successfully tackled many Atari games with complex visual input [21]. Replacing the post-convolutional fully connected layer of DQN by a recurrent LSTM layer allows it to deal with partial observaiblity [10]. However, compared with planning, this approach fails to exploit the underlying sequential nature of decision-making.

We introduce *QMDP-net*, a neural network architecture for planning under partial observability. QMDP-net combines the strengths of model-free learning and model-based planning. A QMDP-net is a recurrent policy network, but it represents a policy by connecting a POMDP model with an algorithm that solves the model, thus embedding the solution structure of planning in a network

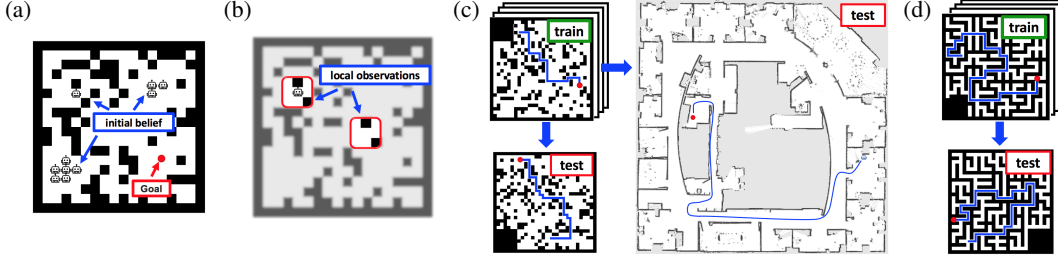

Fig. 1: A robot learning to navigate in partially observable grid worlds. (a) The robot has a map. It has a belief over the initial state, but does not know the exact initial state. (b) Local observations are ambiguous and are insufficient to determine the exact state. (c, d) A policy trained on expert demonstrations in a set of randomly generated environments generalizes to a new environment. It also "transfers" to a much larger real-life environment, represented as a LIDAR map [12].

learning architecture. Specifically, our network uses QMDP [18], a simple, but fast approximate POMDP algorithm, though other more sophisticated POMDP algorithms could be used as well.

A QMDP-net consists of two main network modules (Fig. 2). One represents a Bayesian filter, which integrates the history of an agent's actions and observations into a *belief*, i.e. a probabilistic estimate of the agent's state. The other represents the QMDP algorithm, which chooses the action given the current belief. Both modules are differentiable, allowing the entire network to be trained end-to-end.

We train a QMDP-net on expert demonstrations in a set of randomly generated environments. The trained policy generalizes to new environments and also "transfers" to more complex environments (Fig. 1c–d). Preliminary experiments show that QMDP-net outperformed state-of-the-art network architectures on several robotic tasks in simulation. It successfully solved difficult POMDPs that require reasoning over many time steps, such as the well-known Hallway2 domain [18]. Interestingly, while QMDP-net encodes the QMDP algorithm, it sometimes outperformed the QMDP algorithm in our experiments, as a result of end-to-end learning.

## 2 Background

### 2.1 Planning under Uncertainty

A POMDP is formally defined as a tuple $(S, A, O, T, Z, R)$, where $S$, $A$ and $O$ are the state, action, and observation space, respectively. The state-transition function $T(s, a, s') = P(s'|s, a)$ defines the probability of the agent being in state $s'$ after taking action $a$ in state $s$. The observation function $Z(s, a, o) = p(o|s, a)$ defines the probability of receiving observation $o$ after taking action $a$ in state $s$. The reward function $R(s, a)$ defines the immediate reward for taking action $a$ in state $s$.

In a partially observable world, the agent does not know its exact state. It maintains a *belief*, which is a probability distribution over $S$. The agent starts with an initial belief $b_0$ and updates the belief $b_t$ at each time step $t$ with a Bayesian filter:

$$b_t(s') = \tau(b_{t-1}, a_t, o_t) = \eta Z(s', a_t, o_t) \sum_{s \in S} T(s, a_t, s') b_{t-1}(s), \tag{1}$$

where $\eta$ is a normalizing constant. The belief $b_t$ recursively integrates information from the *entire* past history $(a_1, o_1, a_2, o_2, \ldots, a_t, o_t)$ for decision making. POMDP planning seeks a *policy* $\pi$ that maximizes the *value*, i.e., the expected total discounted reward:

$$V_\pi(b_0) = \mathbb{E}\big(\sum_{t=0}^{\infty} \gamma^t R(s_t, a_{t+1}) \mid b_0, \pi\big), \tag{2}$$

where $s_t$ is the state at time $t$, $a_{t+1} = \pi(b_t)$ is the action that the policy $\pi$ chooses at time $t$, and $\gamma \in (0, 1)$ is a discount factor.

### 2.2 Related Work

To learn policies for decision making in partially observable domains, one approach is to learn models [6, 19, 26] and solve the models through planning. An alternative is to learn policies directly [2, 5]. Model learning is usually not end-to-end. While policy learning can be end-to-end, it does not exploit model information for effective generalization. Our proposed approach combines model-based and

model-free learning by embedding a model and a planning algorithm in a recurrent neural network (RNN) that represents a policy and then training the network end-to-end.

RNNs have been used earlier for learning in partially observable domains [4, 10, 11]. In particular, Hausknecht and Stone extended DQN [21], a convolutional neural network (CNN), by replacing its post-convolutional fully connected layer with a recurrent LSTM layer [10]. Similarly, Mirowski et al. [20] considered learning to navigate in partially observable 3-D mazes. The learned policy generalizes over different goals, but in a fixed environment. Instead of using the generic LSTM, our approach embeds algorithmic structure specific to sequential decision making in the network architecture and aims to learn a policy that generalizes to new environments.

The idea of embedding specific computation structures in the neural network architecture has been gaining attention recently. Tamar et al. implemented value iteration in a neural network, called Value Iteration Network (VIN), to solve Markov decision processes (MDPs) in fully observable domains, where an agent knows its exact state and does not require filtering [34]. Okada et al. addressed a related problem of path integral optimal control, which allows for continuous states and actions [23]. Neither addresses the issue of partial observability, which drastically increases the computational complexity of decision making [24]. Haarnoja et al. [9] and Jonschkowski and Brock [15] developed end-to-end trainable Bayesian filters for probabilistic state estimation. Silver et al. introduced Predictron for value estimation in Markov reward processes [31]. They do not deal with decision making or planning. Both Shankar et al. [28] and Gupta et al. [8] addressed planning under partial observability. The former focuses on learning a model rather than a policy. The learned model is trained on a fixed environment and does not generalize to new ones. The latter proposes a network learning approach to robot navigation in an unknown environment, with a focus on mapping. Its network architecture contains a hierarchical extension of VIN for planning and thus does not deal with partial observability during planning. The QMDP-net extends the prior work on network architectures for MDP planning and for Bayesian filtering. It imposes the POMDP model and computation structure priors on the entire network architecture for planning under partial observability.

## 3   Overview

We want to learn a policy that enables an agent to act effectively in a diverse set of partially observable stochastic environments. Consider, for example, the robot navigation domain in Fig. 1. The environments may correspond to different buildings. The robot agent does not observe its own location directly, but estimates it based on noisy readings from a laser range finder. It has access to building maps, but does not have models of its own dynamics and sensors. While the buildings may differ significantly in their layouts, the underlying reasoning required for effective navigation is similar in all buildings. After training the robot in a few buildings, we want to place the robot in a new building and have it navigate effectively to a specified goal.

Formally, the agent learns a policy for a parameterized set of tasks in partially observable stochastic environments: $\mathcal{W}_\Theta = \{W(\boldsymbol{\theta}) \mid \boldsymbol{\theta} \in \Theta\}$, where $\Theta$ is the set of all parameter values. The parameter value $\boldsymbol{\theta}$ captures a wide variety of task characteristics that vary within the set, including environments, goals, and agents. In our robot navigation example, $\boldsymbol{\theta}$ encodes a map of the environment, a goal, and a belief over the robot's initial state. We assume that all tasks in $\mathcal{W}_\Theta$ share the same state space, action space, and observation space. The agent does not have prior models of its own dynamics, sensors, or task objectives. After training on tasks for some subset of values in $\Theta$, the agent learns a policy that solves $W(\boldsymbol{\theta})$ for any given $\boldsymbol{\theta} \in \Theta$.

A key issue is a general representation of a policy for $\mathcal{W}_\Theta$, without knowing the specifics of $\mathcal{W}_\Theta$ or its parametrization. We introduce the QMDP-net, a recurrent policy network. A QMDP-net represents a policy by connecting a parameterized POMDP model with an approximate POMDP algorithm and embedding both in a single, differentiable neural network. Embedding the model allows the policy to generalize over $\mathcal{W}_\Theta$ effectively. Embedding the algorithm allows us to train the entire network end-to-end and learn a model that compensates for the limitations of the approximate algorithm.

Let $M(\boldsymbol{\theta}) = (S, A, O, f_T(\cdot|\boldsymbol{\theta}), f_Z(\cdot|\boldsymbol{\theta}), f_R(\cdot|\boldsymbol{\theta}))$ be the embedded POMDP model, where $S, A$ and $O$ are the shared state space, action space, observation space designed manually for all tasks in $\mathcal{W}_\Theta$ and $f_T(\cdot|\cdot), f_Z(\cdot|\cdot), f_R(\cdot|\cdot)$ are the state-transition, observation, and reward functions to be learned from data. It may appear that a perfect answer to our learning problem would have

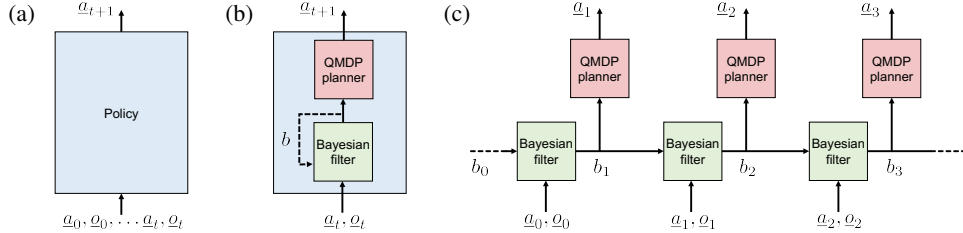

Fig. 2: QMDP-net architecture. (a) A policy maps a history of actions and observations to a new action. (b) A QMDP-net is an RNN that imposes structure priors for sequential decision making under partial observability. It embeds a Bayesian filter and the QMDP algorithm in the network. The hidden state of the RNN encodes the belief for POMDP planning. (c) A QMDP-net unfolded in time.

$f_T(\cdot|\boldsymbol{\theta}), f_Z(\cdot|\boldsymbol{\theta})$, and $f_R(\cdot|\boldsymbol{\theta})$ represent the "true" underlying models of dynamics, observation, and reward for the task $W(\boldsymbol{\theta})$. This is true only if the embedded POMDP algorithm is exact, but not true in general. The agent may learn an alternative model to mitigate an approximate algorithm's limitations and obtain an overall better policy. In this sense, while QMDP-net embeds a POMDP model in the network architecture, it aims to learn a good policy rather than a "correct" model.

A QMDP-net consists of two modules (Fig. 2). One encodes a Bayesian filter, which performs state estimation by integrating the past history of agent actions and observations into a belief. The other encodes QMDP, a simple, but fast approximate POMDP planner [18]. QMDP chooses the agent's actions by solving the corresponding fully observable Markov decision process (MDP) and performing one-step look-ahead search on the MDP values weighted by the belief.

We evaluate the proposed network architecture in an imitation learning setting. We train on a set of expert trajectories with randomly chosen task parameter values in $\Theta$ and test with new parameter values. An expert trajectory consist of a sequence of demonstrated actions and observations $(a_1, o_1, a_2, o_2, \ldots)$ for some $\boldsymbol{\theta} \in \Theta$. The agent does not access the ground-truth states or beliefs along the trajectory during the training. We define loss as the cross entropy between predicted and demonstrated action sequences and use RMSProp [35] for training. See Appendix C.7 for details. Our implementation in Tensorflow [1] is available online at http://github.com/AdaCompNUS/qmdp-net.

## 4 QMDP-Net

We assume that all tasks in a parameterized set $\mathcal{W}_\Theta$ share the same underlying state space $\underline{S}$, action space $\underline{A}$, and observation space $\underline{O}$. We want to learn a QMDP-net policy for $\mathcal{W}_\Theta$, conditioned on the parameters $\boldsymbol{\theta} \in \Theta$. A QMDP-net is a recurrent policy network. The inputs to a QMDP-net are the action $\underline{a}_t \in \underline{A}$ and the observation $\underline{o}_t \in \underline{O}$ at time step $t$, as well as the task parameter $\boldsymbol{\theta} \in \Theta$. The output is the action $\underline{a}_{t+1}$ for time step $t+1$.

A QMDP-net encodes a parameterized POMDP model $M(\boldsymbol{\theta}) = (S, A, O, T = f_T(\cdot|\boldsymbol{\theta}), Z = f_Z(\cdot|\boldsymbol{\theta}), R = f_R(\cdot|\boldsymbol{\theta}))$ and the QMDP algorithm, which selects actions by solving the model approximately. We choose $S$, $A$, and $O$ of $M(\boldsymbol{\theta})$ manually, based on prior knowledge on $\mathcal{W}_\Theta$, specifically, prior knowledge on $\underline{S}$, $\underline{A}$, and $\underline{O}$. In general, $S \neq \underline{S}$, $A \neq \underline{A}$, and $O \neq \underline{O}$. The model states, actions, and observations may be abstractions of their real-world counterparts in the task. In our robot navigation example (Fig. 1), while the robot moves in a continuous space, we choose $S$ to be a grid of finite size. We can do the same for $A$ and $O$, in order to reduce representational and computational complexity. The transition function $T$, observation function $Z$, and reward function $R$ of $M(\boldsymbol{\theta})$ are conditioned on $\boldsymbol{\theta}$, and are learned from data through end-to-end training. In this work, we assume that $T$ is the same for all tasks in $\mathcal{W}_\Theta$ to simplify the network architecture. In other words, $T$ does not depend on $\boldsymbol{\theta}$.

End-to-end training is feasible, because a QMDP-net encodes both a model and the associated algorithm in a single, fully differentiable neural network. The main idea for embedding the algorithm in a neural network is to represent linear operations, such as matrix multiplication and summation, by convolutional layers and represent maximum operations by max-pooling layers. Below we provide some details on the QMDP-net's architecture, which consists of two modules, a filter and a planner.

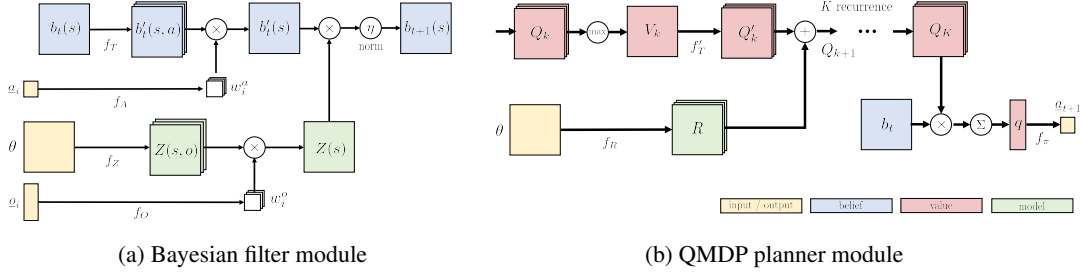

(a) Bayesian filter module          (b) QMDP planner module

Fig. 3: A QMDP-net consists of two modules. (a) The Bayesian filter module incorporates the current action $\underline{a}_t$ and observation $\underline{o}_t$ into the belief. (b) The QMDP planner module selects the action according to the current belief $b_t$.

**Filter module.** The filter module (Fig. 3a) implements a Bayesian filter. It maps from a belief, action, and observation to a next belief, $b_{t+1} = f(b_t | \underline{a}_t, \underline{o}_t)$. The belief is updated in two steps. The first accounts for actions, the second for observations:

$$b'_t(s) = \sum_{s' \in S} T(s', \underline{a}_t, s) b_t(s'), \tag{3}$$

$$b_{t+1}(s) = \eta Z(s, \underline{o}_t) b'_t(s), \tag{4}$$

where $\underline{o}_t \in \underline{O}$ is the observation received after taking action $\underline{a}_t \in \underline{A}$ and $\eta$ is a normalization factor.

We implement the Bayesian filter by transforming Eq. (3) and Eq. (4) to layers of a neural network. For ease of discussion consider our $N{\times}N$ grid navigation task (Fig. 1a–c). The agent does not know its own state and only observes neighboring cells. It has access to the task parameter $\boldsymbol{\theta}$ that encodes the obstacles, goal, and a belief over initial states. Given the task, we choose $M(\boldsymbol{\theta})$ to have a $N{\times}N$ state space. The belief, $b_t(s)$, is now an $N{\times}N$ tensor.

Eq. (3) is implemented as a convolutional layer with $|A|$ convolutional filters. We denote the convolutional layer by $f_T$. The kernel weights of $f_T$ encode the transition function $T$ in $M(\boldsymbol{\theta})$. The output of the convolutional layer, $b'_t(s, a)$, is a $N{\times}N{\times}|A|$ tensor.

$b'_t(s, a)$ encodes the updated belief after taking each of the actions, $a \in A$. We need to select the belief corresponding to the last action taken by the agent, $\underline{a}_t$. We can directly index $b'_t(s, a)$ by $\underline{a}_t$ if $A = \underline{A}$. In general $A \neq \underline{A}$, so we cannot use simple indexing. Instead, we will use "soft indexing". First we encode actions in $\underline{A}$ to actions in $A$ through a learned function $f_A$. $f_A$ maps from $\underline{a}_t$ to an indexing vector $w_t^a$, a distribution over actions in $A$. We then weight $b'_t(s, a)$ by $w_t^a$ along the appropriate dimension, i.e.

$$b'_t(s) = \sum_{a \in A} b'_t(s, a) w_t^a. \tag{5}$$

Eq. (4) incorporates observations through an observation model $Z(s, o)$. Now $Z(s, o)$ is a $N{\times}N{\times}|O|$ tensor that represents the probability of receiving observation $o \in O$ in state $s \in S$. In our grid navigation task observations depend on the obstacle locations. We condition $Z$ on the task parameter, $Z(s, o) = f_Z(s, o | \boldsymbol{\theta})$ for $\boldsymbol{\theta} \in \Theta$. The function $f_Z$ is a neural network, mapping from $\boldsymbol{\theta}$ to $Z(s, o)$. In this paper $f_Z$ is a CNN.

$Z(s, o)$ encodes observation probabilities for each of the observations, $o \in O$. We need the observation probabilities for the last observation $\underline{o}_t$. In general $O \neq \underline{O}$ and we cannot index $Z(s, o)$ directly. Instead, we will use soft indexing again. We encode observations in $\underline{O}$ to observations in $O$ through $f_O$. $f_O$ is a function mapping from $\underline{o}_t$ to an indexing vector, $w_t^o$, a distribution over $O$. We then weight $Z(s, o)$ by $w_t^o$, i.e.

$$Z(s) = \sum_{o \in O} Z(s, o) w_t^o. \tag{6}$$

Finally, we obtain the updated belief, $b_{t+1}(s)$, by multiplying $b'_t(s)$ and $Z(s)$ element-wise, and normalizing over states. In our setting the initial belief for the task $W(\boldsymbol{\theta})$ is encoded in $\boldsymbol{\theta}$. We initialize the belief in QMDP-net through an additional encoding function, $b_0 = f_B(\boldsymbol{\theta})$.

**Planner module.** The QMDP planner (Fig. 3b) performs value iteration at its core. $Q$ values are computed by iteratively applying Bellman updates,

$$Q_{k+1}(s,a) = R(s,a) + \gamma \sum_{s' \in S} T(s,a,s') V_k(s'),\qquad(7)$$

$$V_k(s) = \max_a Q_k(s,a).\qquad(8)$$

Actions are then selected by weighting the $Q$ values with the belief.

We can implement value iteration using convolutional and max pooling layers [28, 34]. In our grid navigation task $Q(s,a)$ is a $N \times N \times |A|$ tensor. Eq. (8) is expressed by a max pooling layer, where $Q_k(s,a)$ is the input and $V_k(s)$ is the output. Eq. (7) is a $N \times N$ convolution with $|A|$ convolutional filters, followed by an addition operation with $R(s,a)$, the reward tensor. We denote the convolutional layer by $f'_T$. The kernel weights of $f'_T$ encode the transition function $T$, similarly to $f_T$ in the filter. Rewards for a navigation task depend on the goal and obstacles. We condition rewards on the task parameter, $R(s,a) = f_R(s,a|\boldsymbol{\theta})$. $f_R$ maps from $\boldsymbol{\theta}$ to $R(s,a)$. In this paper $f_R$ is a CNN.

We implement $K$ iterations of Bellman updates by stacking the layers representing Eq. (7) and Eq. (8) $K$ times with tied weights. After $K$ iterations we get $Q_K(s,a)$, the approximate $Q$ values for each state-action pair. We weight the $Q$ values by the belief to obtain action values,

$$q(a) = \sum_{s \in S} Q_K(s,a) b_t(s).\qquad(9)$$

Finally, we choose the output action through a low-level policy function, $f_\pi$, mapping from $q(a)$ to the action output, $\underline{a}_{t+1}$.

QMDP-net naturally extends to higher dimensional discrete state spaces (e.g. our maze navigation task) where $n$-dimensional convolutions can be used [14]. While $M(\boldsymbol{\theta})$ is restricted to a discrete space, we can handle continuous tasks $\mathcal{W}_\Theta$ by simultaneously learning a discrete $M(\boldsymbol{\theta})$ for planning, and $f_A, f_O, f_B, f_\pi$ to map between states, actions and observations in $\mathcal{W}_\Theta$ and $M(\boldsymbol{\theta})$.

## 5 Experiments

The main objective of the experiments is to understand the benefits of structure priors on learning neural-network policies. We create several alternative network architectures by gradually relaxing the structure priors and evaluate the architectures on simulated robot navigation and manipulation tasks. While these tasks are simpler than, for example, Atari games, in terms of visual perception, they are in fact very challenging, because of the sophisticated long-term reasoning required to handle partial observability and distant future rewards. Since the exact state of the robot is unknown, a successful policy must reason over many steps to gather information and improve state estimation through partial and noisy observations. It also must reason about the trade-off between the cost of information gathering and the reward in the distance future.

### 5.1 Experimental Setup

We compare the QMDP-net with a number of related alternative architectures. Two are QMDP-net variants. *Untied QMDP-net* relaxes the constraints on the planning module by untying the weights representing the state-transition function over the different CNN layers. *LSTM QMDP-net* replaces the filter module with a generic LSTM module. The other two architectures do not embed POMDP structure priors at all. *CNN+LSTM* is a state-of-the-art deep CNN connected to an LSTM. It is similar to the DRQN architecture proposed for reinforcement learning under partially observability [10]. *RNN* is a basic recurrent neural network with a single fully-connected hidden layer. RNN contains no structure specific to planning under partial observability.

Each experimental domain contains a parameterized set of tasks $\mathcal{W}_\Theta$. The parameters $\boldsymbol{\theta}$ encode an environment, a goal, and a belief over the robot's initial state. To train a policy for $\mathcal{W}_\Theta$, we generate random environments, goals, and initial beliefs. We construct ground-truth POMDP models for the generated data and apply the QMDP algorithm. If the QMDP algorithm successfully reaches the goal, we then retain the resulting sequence of action and observations $(a_1, o_1, a_2, o_2, \dots)$ as an expert trajectory, together with the corresponding environment, goal, and initial belief. It is important to note that the ground-truth POMDPs are used only for generating expert trajectories and not for learning the QMDP-net.

For fair comparison, we train all networks using the same set of expert trajectories in each domain. We perform basic search over training parameters, the number of layers, and the number of hidden units for each network architecture. Below we briefly describe the experimental domains. See Appendix C for implementation details.

**Grid-world navigation.** A robot navigates in an unknown building given a floor map and a goal. The robot is uncertain of its own location. It is equipped with a LIDAR that detects obstacles in its direct neighborhood. The world is uncertain: the robot may fail to execute desired actions, possibly because of wheel slippage, and the LIDAR may produce false readings. We implemented a simplified version of this task in a discrete $n \times n$ grid world (Fig. 1c). The task parameter $\boldsymbol{\theta}$ is represented as an $n \times n$ image with three channels. The first channel encodes the obstacles in the environment, the second channel encodes the goal, and the last channel encodes the belief over the robot's initial state. The robot's state represents its position in the grid. It has five actions: moving in each of the four canonical directions or staying put. The LIDAR observations are compressed into four binary values corresponding to obstacles in the four neighboring cells. We consider both a deterministic and a stochastic variant of the domain. The stochastic variant adds action and observation uncertainties. The robot fails to execute the specified move action and stays in place with probability $0.2$. The observations are faulty with probability $0.1$ independently in each direction. We trained a policy using expert trajectories from $10,000$ random environments, $5$ trajectories from each environment. We then tested on a separate set of $500$ random environments.

**Maze navigation.** A differential-drive robot navigates in a maze with the help of a map, but it does not know its pose (Fig. 1d). This domain is similar to the grid-world navigation, but it is significant more challenging. The robot's state contains both its position and orientation. The robot cannot move freely because of kinematic constraints. It has four actions: move forward, turn left, turn right and stay put. The observations are relative to the robot's current orientation, and the increased ambiguity makes it more difficult to localize the robot, especially when the initial state is highly uncertain. Finally, successful trajectories in mazes are typically much longer than those in randomly-generated grid worlds. Again we trained on expert trajectories in $10,000$ randomly generated mazes and tested them in $500$ new ones.

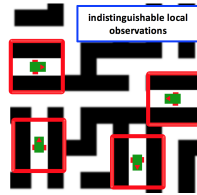

Fig. 4: Highly ambiguous observations in a maze. The four observations (in red) are the same, despite that the robot states are all different.

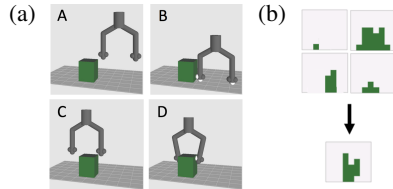

Fig. 5: Object grasping using touch sensing. (a) An example [3]. (b) Simplified 2-D object grasping. Objects from the training set (top) and the test set (bottom).

**2-D object grasping.** A robot gripper picks up novel objects from a table using a two-finger hand with noisy touch sensors at the finger tips. The gripper uses the fingers to perform compliant motions while maintaining contact with the object or to grasp the object. It knows the shape of the object to be grasped, maybe from an object database. However, it does not know its own pose relative to the object and relies on the touch sensors to localize itself. We implemented a simplified 2-D variant of this task, modeled as a POMDP [13]. The task parameter $\boldsymbol{\theta}$ is an image with three channels encoding the object shape, the grasp point, and a belief over the gripper's initial pose. The gripper has four actions, each moving in a canonical direction unless it touches the object or the environment boundary. Each finger has $3$ binary touch sensors at the tip, resulting in $64$ distinct observations. We trained on expert demonstration on $20$ different objects with $500$ randomly sampled poses for each object. We then tested on $10$ previously unseen objects in random poses.

### 5.2  Choosing QMDP-Net Components for a Task

Given a new task $\mathcal{W}_{\Theta}$, we need to choose an appropriate neural network representation for $M(\boldsymbol{\theta})$. More specifically, we need to choose $S, A$ and $O$, and a representation for the functions $f_R, f_T, f'_T, f_Z, f_O, f_A, f_B, f_\pi$. This provides an opportunity to incorporate domain knowledge in a principled way. For example, if $\mathcal{W}_{\Theta}$ has a local and spatially invariant connectivity structure, we can choose convolutions with small kernels to represent $f_T$, $f_R$ and $f_Z$.

In our experiments we use $S = N \times N$ for $N \times N$ grid navigation, and $S = N \times N \times 4$ for $N \times N$ maze navigation where the robot has 4 possible orientations. We use $|A| = |\underline{A}|$ and $|O| = |\underline{O}|$ for all tasks except for the object grasping task, where $|\underline{O}| = 64$ and $|O| = 16$. We represent $f_T$, $f_R$ and $f_Z$ by CNN components with $3 \times 3$ and $5 \times 5$ kernels depending on the task. We enforce that $f_T$ and $f_Z$ are proper probability distributions by using softmax and sigmoid activations on the convolutional kernels, respectively. Finally, $f_O$ is a small fully connected component, $f_A$ is a one-hot encoding function, $f_\pi$ is a single softmax layer, and $f_B$ is the identity function.

We can adjust the *amount of planning* in a QMDP-net by setting $K$. A large $K$ allows propagating information to more distant states without affecting the number of parameters to learn. However, it results in deeper networks that are computationally expensive to evaluate and more difficult to train. We used $K = 20 \ldots 116$ depending on the problem size. We were able to transfer policies to larger environments by increasing $K$ up to $450$ when executing the policy.

In our experiments the representation of the task parameter $\boldsymbol{\theta}$ is isomorphic to the chosen state space $S$. While the architecture is not restricted to this setting, we rely on it to represent $f_T, f_Z, f_R$ by convolutions with small kernels. Experiments with a more general class of problems is an interesting direction for future work.

### 5.3 Results and Discussion

The main results are reported in Table 1. Some additional results are reported in Appendix A. For each domain, we report the task success rate and the average number of time steps for task completion. Comparing the completion time is meaningful only when the success rates are similar.

**QMDP-net successfully learns policies that generalize to new environments.** When evaluated on new environments, the QMDP-net has higher success rate and faster completion time than the alternatives in nearly all domains. To understand better the performance difference, we specifically compared the architectures in a fixed environment for navigation. Here only the initial state and the goal vary across the task instances, while the environment remains the same. See the results in the last row of Table 1. The QMDP-net and the alternatives have comparable performance. Even RNN performs very well. Why? In a fixed environment, a network may learn the features of an optimal policy directly, e.g., going straight towards the goal. In contrast, the QMDP-net learns a model for *planning*, i.e., generating a near-optimal policy for a given arbitrary environment.

**POMDP structure priors improve the performance of learning complex policies.** Moving across Table 1 from left to right, we gradually relax the POMDP structure priors on the network architecture. As the structure priors weaken, so does the overall performance. However, strong priors sometimes over-constrain the network and result in degraded performance. For example, we found that tying the weights of $f_T$ in the filter and $f_T'$ in the planner may lead to worse policies. While both $f_T$ and $f_T'$ represent the same underlying transition dynamics, using different weights allows each to choose its own approximation and thus greater flexibility. We shed some light on this issue and visualize the learned POMDP model in Appendix B.

**QMDP-net learns "incorrect", but useful models.** Planning under partial observability is intractable in general, and we must rely on approximation algorithms. A QMDP-net encodes both a POMDP model and QMDP, an approximate POMDP algorithm that solves the model. We then train the network end-to-end. This provides the opportunity to learn an "incorrect", but useful model that compensates the limitation of the approximation algorithm, in a way similar to reward shaping in reinforcement learning [22]. Indeed, our results show that the QMDP-net achieves higher success rate than QMDP in nearly all tasks. In particular, QMDP-net performs well on the well-known Hallway2 domain, which is designed to expose the weakness of QMDP resulting from its myopic planning horizon. The planning algorithm is the same for both the QMDP-net and QMDP, but the QMDP-net learns a more effective model from expert demonstrations. This is true even though QMDP generates the expert data for training. We note that the expert data contain only successful QMDP demonstrations. When both successful and unsuccessful QMDP demonstrations were used for training, the QMDP-net did not perform better than QMDP, as one would expect.

**QMDP-net policies learned in small environments transfer directly to larger environments.** Learning a policy for large environments from scratch is often difficult. A more scalable approach

Table 1: Performance comparison of QMDP-net and alternative architectures for recurrent policy networks. SR is the success rate in percentage. Time is the average number of time steps for task completion. D-$n$ and S-$n$ denote deterministic and stochastic variants of a domain with environment size $n \times n$.

| Domain | QMDP | | QMDP-net | | Untied QMDP-net | | LSTM QMDP-net | | CNN +LSTM | | RNN | |
|---|---|---|---|---|---|---|---|---|---|---|---|---|
| | SR | Time | SR | Time | SR | Time | SR | Time | SR | Time | SR | Time |
| Grid D-10 | 99.8 | 8.8 | 99.6 | 8.2 | 98.6 | 8.3 | 84.4 | 12.8 | 90.0 | 13.4 | 87.8 | 13.4 |
| Grid D-18 | 99.0 | 15.5 | 99.0 | 14.6 | 98.8 | 14.8 | 43.8 | 27.9 | 57.8 | 33.7 | 35.8 | 24.5 |
| Grid D-30 | 97.6 | 24.6 | 98.6 | 25.0 | 98.8 | 23.9 | 22.2 | 51.1 | 19.4 | 45.2 | 16.4 | 39.3 |
| Grid S-18 | 98.1 | 23.9 | 98.8 | 23.9 | 95.9 | 24.0 | 23.8 | 55.6 | 41.4 | 65.9 | 34.0 | 64.1 |
| Maze D-29 | 63.2 | 54.1 | 98.0 | 56.5 | 95.4 | 62.5 | 9.8 | 57.2 | 9.2 | 41.4 | 9.8 | 47.0 |
| Maze S-19 | 63.1 | 50.5 | 93.9 | 60.4 | 98.7 | 57.1 | 18.9 | 79.0 | 19.2 | 80.8 | 19.6 | 82.1 |
| Hallway2 | 37.3 | 28.2 | 82.9 | 64.4 | 69.6 | 104.4 | 82.8 | 89.7 | 77.8 | 99.5 | 68.0 | 108.8 |
| Grasp | 98.3 | 14.6 | 99.6 | 18.2 | 98.9 | 20.4 | 91.4 | 26.4 | 92.8 | 22.1 | 94.1 | 25.7 |
| Intel Lab | 90.2 | 85.4 | 94.4 | 107.7 | 20.0 | 55.3 | - | | - | | - | |
| Freiburg | 88.4 | 66.9 | 93.2 | 81.1 | 37.4 | 51.7 | - | | - | | - | |
| Fixed grid | 98.8 | 17.4 | 98.6 | 17.6 | 99.8 | 17.0 | 97.0 | 19.7 | 98.4 | 19.9 | 98.0 | 19.8 |

would be to learn a policy in small environments and transfer it to large environments by repeating the reasoning process. To transfer a learned QMDP-net policy, we simply expand its planning module by adding more recurrent layers. Specifically, we trained a policy in randomly generated $30 \times 30$ grid worlds with $K = 90$. We then set $K = 450$ and applied the learned policy to several real-life environments, including Intel Lab ($100 \times 101$) and Freiburg ($139 \times 57$), using their LIDAR maps (Fig. 1c) from the Robotics Data Set Repository [12]. See the results for these two environments in Table 1. Additional results with different $K$ settings and other buildings are available in Appendix A.

# 6  Conclusion

A QMDP-net is a deep recurrent policy network that embeds POMDP structure priors for planning under partial observability. While generic neural networks learn a direct mapping from inputs to outputs, QMDP-net learns how to *model* and *solve* a planning task. The network is fully differentiable and allows for end-to-end training.

Experiments on several simulated robotic tasks show that learned QMDP-net policies successfully generalize to new environments and transfer to larger environments as well. The POMDP structure priors and end-to-end training substantially improve the performance of learned policies. Interestingly, while a QMDP-net encodes the QMDP algorithm for planning, learned QMDP-net policies sometimes outperform QMDP.

There are many exciting directions for future exploration. First, a major limitation of our current approach is the state space representation. The value iteration algorithm used in QMDP iterates through the entire state space and is well known to suffer from the "curse of dimensionality". To alleviate this difficulty, the QMDP-net, through end-to-end training, may learn a much smaller abstract state space representation for planning. One may also incorporate hierarchical planning [8]. Second, QMDP makes strong approximations in order to reduce computational complexity. We want to explore the possibility of embedding more sophisticated POMDP algorithms in the network architecture. While these algorithms provide stronger planning performance, their algorithmic sophistication increases the difficulty of learning. Finally, we have so far restricted the work to imitation learning. It would be exciting to extend it to reinforcement learning. Based on earlier work [28, 34], this is indeed promising.

**Acknowledgments**  We thank Leslie Kaelbling and Tomás Lozano-Pérez for insightful discussions that helped to improve our understanding of the problem. The work is supported in part by Singapore Ministry of Education AcRF grant MOE2016-T2-2-068 and National University of Singapore AcRF grant R-252-000-587-112.

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
