[Supplementary Material · QMDP-Net appendix.pdf]

# A  Supplementary Experiments

## A.1   Navigation on Large LIDAR Maps

We provide results on additional environments for the LIDAR map navigation task. LIDAR maps are obtained from [33]. See Section C.5 for details. **Intel** corresponds to Intel Research Lab. **Freiburg** corresponds to Freiburg, Building 079. **Belgioioso** corresponds to Belgioioso Castle. **MIT** corresponds to the western wing of the MIT CSAIL building. We note the size of the grid size $NxM$ for each environment. A QMDP-net policy is trained on the $30x30$-D grid navigation domain on randomly generated environments using $K = 90$. We then execute the learned QMDP-net policy with different $K$ settings, i.e. we add convolutional layers to the planner that share the same kernel weights. We report the task success rate and the average number of time steps for task completion.

Table 2: Additional results for navigation on large LIDAR maps.

| Domain | QMDP SR | QMDP Time | QMDP-net K=450 SR | QMDP-net K=450 Time | QMDP-net K=180 SR | QMDP-net K=180 Time | QMDP-net K=90 SR | QMDP-net K=90 Time | Untied QMDP-net SR | Untied QMDP-net Time |
|---|---|---|---|---|---|---|---|---|---|---|
| Intel $100{\times}101$ | 90.2 | 85.4 | **94.4** | 108.0 | 83.4 | 89.6 | 40.8 | 78.6 | 20.0 | 55.3 |
| Freiburg $139{\times}57$ | 88.4 | 66.9 | 92.0 | 91.4 | **93.2** | 81.1 | 55.8 | 68.0 | 37.4 | 51.7 |
| Belgioioso $151{\times}35$ | 95.8 | 63.9 | **95.4** | 71.8 | 90.6 | 62.0 | 60.0 | 54.3 | 41.0 | 47.7 |
| MIT $41{\times}83$ | 94.4 | 42.6 | 91.4 | 53.8 | **96.0** | 48.5 | 86.2 | 45.4 | 66.6 | 41.4 |

In the conventional setting, when value iteration is executed on a fully known MDP, increasing $K$ improves the value function approximation and improves the policy in return for the increased computation. In a QMDP-net increasing $K$ has two effects on the overall planning quality. Estimation accuracy of the latent values increases and reward information can propagate to more distant states. On the other hand the learned latent model does not necessarily fit the true underlying model, and it can be overfitted to the $K$ setting during training. Therefore a too high $K$ can degrade the overall performance. We found that $K_{test} = 2K_{train}$ significantly improved success rates in all our test cases. Further increasing $K_{test} = 5K_{train}$ was beneficial in the Intel and Belgioioso environments, but it slightly decreased success rates for the Freiburg and MIT environments.

We compare QMDP-net to its untied variant, Untied QMDP-net. We cannot expand the layers of Untied QMDP-net during execution. In consequence, the performance is poor. Note that the other alternative architectures we considered are specific to the input size and thus they are not applicable.

## A.2   Learning "Incorrect" but Useful Models

We demonstrate that an "incorrect" model can result in better policies when solved by the approximate QMDP algorithm. We compute QMDP policies on a POMDP with modified reward values, then evaluate the policies using the original rewards. We use the deterministic $29{\times}29$ maze navigation task where QMDP did poorly. We attempt to shape rewards manually. Our motivation is to break symmetry in the model, and to implicitly encourage information gathering and compensate for the one-step look-ahead approximation in QMDP. **Modified 1.** We increase the cost for the stay actions to $20$ times of its original value. **Modified 2.** We increase the cost for the stay action to $50$ times of its original value, and the cost for the turn right action to $10$ times of its original value.

Table 3: QMDP policies computed on an "incorrect" model and evaluated on the "correct" model.

| Variant | SR | Time | Original reward |
|---|---|---|---|
| Original | 63.2 | 54.1 | 1.09 |
| Modified 1 | 65.0 | 58.1 | 1.71 |
| Modified 2 | **93.0** | 71.4 | 4.96 |

Why does the "correct" model result in poor policies when solved by QMDP? At a given point the $Q$ value for a set of possible states may be high for the turn left action and low for the turn right action;

while for another set of states it may be the opposite way around. In expectation, both next states have lower value than the current one, thus the policy chooses the stay action, the robot does not gather information and it is stuck in one place. Results demonstrate that planning on an "incorrect" model may improve the performance on the "correct" model.

# B Visualizing the Learned Model

## B.1 Value Function

We plot the value function predicted by a QMDP-net for the $18 \times 18$ stochastic grid navigation task. We used $K = 54$ iterations in the QMDP-net. As one would expect, states close to the goal have high values.

Fig. 6: Map of a test environment and the corresponding learned value function $V_K$.

## B.2 Belief Propagation

We plot the execution of a learned QMDP-net policy and the internal belief propagation on the $18 \times 18$ stochastic grid navigation task. The first row in Fig. 7 shows the environment including the goal (red) and the unobserved pose of the robot (blue). The second row shows ground-truth beliefs for reference. We do not access ground-truth beliefs during training except for the initial belief. The third row shows beliefs predicted by a QMDP-net. The last row shows the difference between the ground-truth and predicted beliefs.

Fig. 7: Policy execution and belief propagation in the $18 \times 18$ stochastic grid navigation task.

The figure demonstrates that QMDP-net was able to learn a reasonable filter for state estimation in a noisy environment. In the depicted example the initial belief is uniform over approximately half of the state space (Step 0). Due to the highly uncertain initial belief and the observation noise the robot stays in place for two steps (Step 1 and 2). After two steps the state estimation is still highly uncertain, but it is mostly spread out right from the goal. Therefore, moving left is a reasonable choice (Step 3). After an additional stay action (Step 4) the belief distribution is small enough and the robot starts moving towards the goal (not shown).

## B.3   State-Transition Function

We plot the learned and ground-truth state-transition functions. Columns of the table correspond to actions. The first row shows the ground-truth transition function. The second row shows $f_T$, the learned state-transition function in the filter. The third row shows $f'_T$, the learned state-transition function in the planner.

Fig. 8: Learned transition function $T$ in the $18 \times 18$ stochastic grid navigation task.

While both $f_T$ and $f'_T$ represent the same underlying transition dynamics, the learned transition probabilities are different in the filter and planner. Different weights allows each module to choose its own approximation and thus provides greater flexibility. The actions in the model $a \in A$ are learned abstractions of the agent's actions $\underline{a} \in \underline{A}$. Indeed, in the planner the learned transition probabilities for action $a_i \in A$ do not match the transition probabilities of $\underline{a}_i \in \underline{A}$.

## B.4   Reward Function

Next plot the learned reward function $R$ for each action $a \in A$.

Fig. 9: Learned reward function $R$ in the $18 \times 18$ stochastic grid navigation domain.

While the learned rewards do not directly correspond to rewards in the underlying task, they are reasonable: obstacles are assigned negative rewards and the goal is assigned a positive reward. Note that learned reward values correspond to the reward *after* taking an action, therefore they should be interpreted together with the corresponding transition probabilities (third row of Fig. 8).

# C Implementation Details

## C.1 Grid-World Navigation

We implement the grid navigation task in randomly generated discrete $N \times N$ grids where each cell has $p = 0.25$ probability of being an obstacle. The robot has 5 actions: move in the four canonical directions and stay put. Observations are four binary values corresponding to obstacles in the four neighboring cells. We consider a deterministic variant (denoted by -D) and a stochastic variant (denoted by -S). In the stochastic variant the robot fails to execute each action with probability $P_t = 0.2$, in which case it stays in place. The observations are faulty with probability $P_o = 0.1$ independently in each direction. Since we receive observations from 4 directions, the probability of receiving the correct observation vector is only $0.9^4 = 0.656$. The task parameter, $\boldsymbol{\theta}$, is an $N \times N \times 3$ image that encodes information about the environment. The first channel encodes obstacles, 1 for obstacles, 0 for free space. The second channel encodes the goal, 1 for the goal, 0 otherwise. The third channel encodes the initial belief over robot states, each pixel value corresponds to the probability of the robot being in the corresponding state.

We construct a ground-truth POMDP model to obtain expert trajectories for training. It is important to note that the learning agent has no access to the ground-truth POMDP models. In the ground-truth model the robot receives a reward of $-0.1$ for each step, $+20$ for reaching the goal, and $-10$ for bumping into an obstacle. We use QMDP to solve the POMDP model, and execute the QMDP policy to obtain expert trajectories. We use $10,000$ random grids for training. Initial and goal states are sampled from the free space uniformly. We exclude samples where there is no feasible path. The initial belief is uniform over a random fraction of the free space which includes the underlying initial state. More specifically, the number of non-zero values in the initial-belief are sampled from $\{1, 2, \ldots N_f/2, N_f\}$ where $N_f$ is the number of free cells in the grid. For each grid we generate 5 expert trajectories with different initial state, initial belief and goal. Note that we do not access the true beliefs after the first step nor the underlying states along the trajectory.

We test on a set of $500$ environments generated separately in equal conditions. We declare failure after $10N$ steps without reaching the goal. Note that the expert policy is sub-optimal and it may fail to reach the goal. We exclude these samples from the training set but include them in the test set.

We choose the structure of $M(\boldsymbol{\theta})$, the model in QMDP-net, to match the structure of the underlying task. The transition function in the filter $f_T$ and the planner $f'_T$ are both $3 \times 3$ convolutions. While they both represent the same transition function we do not tie their weights. We apply a softmax function on the kernel matrix so its values sum to one. The reward function, $f_R$, is a CNN with two convolutional layers. The first has $3 \times 3$ kernel, 150 filters, ReLU activation. The second has $1 \times 1$ kernel, 5 filters and linear activation. The observation model, $f_Z$, is a similar two-layer CNN. The first convolution has a $3 \times 3$ kernel, 150 filters, linear activation. The second has $1 \times 1$ kernel, 17 filters and linear activation. The action mapping, $f_A$, is a one-hot encoding function. The observation mapping, $f_O$, is a fully connected network with one hidden layer with 17 units and tanh activation. It has 17 output units and softmax activation. The low-level policy function, $f_\pi$, is a single softmax layer. The state space mapping function, $f_B$, is the identity function. Finally, we choose the number of iterations in the planner module, $K = \{30, 54, 90\}$ for grids of size $N = \{10, 18, 30\}$ respectively.

The $3 \times 3$ convolutions in $f_T$ and $f_Z$ imply that $T$ and $O$ are spatially invariant and local. In the underlying task the locality assumption holds but spatial invariance does not: transitions depend on the arrangement of obstacles. Nevertheless, the additional flexibility in the model allows QMDP-net to learn high-quality policies, e.g. by shaping the rewards and the observation function.

## C.2 Maze Navigation

In the maze navigation task a differential drive robot has to navigate to a given goal. We generate random mazes on $N \times N$ grids using Kruskal's algorithm. The state space has 3 dimensions where the third dimension represents 4 possible orientations of the robot. The goal configuration is invariant to the orientation. The robot now has 4 actions: move forward, turn left, turn right and stay put. The initial belief is chosen in a similar manner to the grid navigation case but in the 3-D space. The observations are identical to grid navigation but they are relative to the robot's orientation, which significantly increases the difficulty of state estimation. The stochastic variant (denoted by -S) has

a motion and observation noise identical to the grid navigation. Training and test data is prepared identically as well. We use $K = \{76, 116\}$ for mazes of size $N = \{19, 29\}$ respectively.

We use a model in QMDP-net with a 3-dimensional state space of size $N \times N \times 4$ and an action space with 4 actions. The components of the network are chosen identically to the previous case, except that all CNN components operate on 3-D tensors of size $N \times N \times 4$. While it would be possible to use 3-D convolutions, we treat the third dimension as channels of a 2-D image instead, and use conventional 2-D convolutions. If the output of the last convolutional layer is of size $N \times N \times N_c$ for the grid navigation task, it is of size $N \times N \times 4N_c$ for the maze navigation task. When necessary, these tensors are transformed into a 4 dimensional form $N \times N \times 4 \times N_c$ and the max-pool or softmax activation is computed along the last dimension.

## C.3 Object Grasping

We consider a 2-D implementation of the grasping task based on the POMDP model proposed by Hsiao et al. [13]. Hsiao et al. focused on the difficulty of planning with high uncertainty and solved manually designed POMDPs for single objects. We phrase the problem as a learning task where we have no access to a model and we do not know all objects in advance. In our setting the robot receives an image of the target object and a feasible grasp point, but it does not know its pose relative to the object. We aim to learn a policy on a set of object that generalizes to similar but unseen objects.

The object and the gripper are represented in a discrete grid. The workspace is a $14 \times 14$ grid, and the gripper is a "U" shape in the grid. The gripper moves in the four canonical directions, unless it reaches the boundaries of the workspace or it is touching the object. in which case it stays in place. The gripper fails to move with probability $0.2$. The gripper has two fingers with 3 touch sensors on each finger. The touch sensors indicate contact with the object or reaching the limits of the workspace. The sensors produce an incorrect reading with probability $0.1$ independently for each sensor. In each trial an object is placed on the bottom of the workspace at a random location. The initial gripper pose is unknown; the belief over possible states is uniform over a random fraction of the upper half of the workspace. The local observations, $o_t$, are readings from the touch sensors. The task parameter $\boldsymbol{\theta}$ is an image with three channels. The first channel encodes the environment with an object; the second channel encodes the position of the target grasping point; the third channel encodes the initial belief over the gripper position.

We have 30 artificial objects of different sizes up to $6 \times 6$ grid cells. Each object has at least one cell on its top that the gripper can grasp. For training we use 20 of the objects. We generate 500 expert trajectories for each object in random configuration. We test the learned policies on 10 new objects in 20 random configurations each. The expert trajectories are obtained by solving a ground-truth POMDP model by the QMDP algorithm. In the ground-truth POMDP the robot receives a reward of 1 for reaching the grasp point and 0 for every other state.

In QMDP-net we choose a model with $S = 14 \times 14$, $|A| = 4$ and $|O| = 16$. Note that the underlying task has $|O| = 64$ possible observations. The network components are chosen similarly to the grid navigation task, but the first convolution kernel in $f_z$ is increased to $5 \times 5$ to account for more distant observations. We set the number of iterations $K = 20$.

## C.4 Hallway2

The Hallway2 navigation problem was proposed by Littman et al. [18] and has been used as a benchmark problem for POMDP planning [27]. It was specifically designed to expose the weakness of the QMDP algorithm resulting from its myopic planning horizon. While QMDP-net embeds the QMDP algorithm, through end-to-end training QMDP-net was able to learn a model that is significantly more effective given the QMDP algorithm.

Hallway2 is a particular instance of the maze problem that involves more complex dynamics and high noise. For details we refer to the original problem definition [18]. We train a QMDP-net on random $8 \times 8$ grids generated similarly to the grid navigation case, but using transitions that match the Hallway2 POMDP model. We then execute the learned policy on a particularly difficult instance of this problem that embeds the Hallway2 layout in a $8 \times 8$ grid. The initial state is uniform over the full state space. In each trial the robot starts from a random underlying state. The trial is deemed unsuccessful after 251 steps.

## C.5 Navigation on a Large LIDAR Map

We obtain real-world building layouts using 2-D laser data from the Robotics Data Set Repository [12]. More specifically, we use SLAM maps preprocessed to gray-scale images available online [33]. We downscale the raw images to $NxM$ and classify each pixel to be free or an obstacle by simple thresholding. The resulting maps are shown in Fig. 10. We execute policies in simulation where a grid is defined by the preprocessed map. The simulation employs the same dynamics as the grid navigation domain. The initial state and initial belief are chosen identically to the grid navigation case.

Fig. 10: Preprocessed $N \times M$ maps. **A,** Intel Research Lab, $100 \times 101$. **B,** Freiburg, building 079, $139 \times 57$. **C,** Belgioioso Castle, $151 \times 35$. **D,** western wing of the MIT CSAIL building, $41 \times 83$.

A QMDP-net policy is trained on the $30x30$-D grid navigation task on randomly generated environments. For training we set $K = 90$ in the QMDP-net. We then execute the learned policy on the LIDAR maps. To account for the larger grid size we increase the number of iterations to $K = 450$ when executing the policy.

## C.6 Architectures for Comparison

We compare QMDP-net with two of its variants where we remove some of the POMDP priors embedded in the network (Untied QMDP-net, LSTM QMDP-net). We also compare with two generic network architectures that do not embed structural priors for decision making (CNN+LSTM, RNN). We also considered additional architectures for comparison, including networks with GRU [7] and ConvLSTM [36] cells. ConvLSTM is a variant of LSTM where the fully connected layers are replaced by convolutions. These architectures performed worse than CNN+LSTM for most of our task.

**Untied QMDP-net.** We obtain Untied QMDP-net by untying the kernel weights in the convolutional layers that implement value iteration in the planner module of QMDP-net. We also remove the softmax activation on the kernel weights. This is equivalent to allowing a different transition model at each iteration of value iteration, and allowing transition probabilities that do not sum to one. In principle, Untied QMDP-net can represent the same policy as QMDP-net and it has some additional flexibility. However, Untied QMDP-net has more parameters to learn as $K$ increases. The training difficulty increases with more parameters, especially on complex domains or when training with small amount of data.

**LSTM QMDP-net.** In LSTM QMDP-net we replace the filter module of QMDP-net with a generic LSTM network but keep the value iteration implementation in the planner. The output of the LSTM component is a belief estimate which is input to the planner module of QMDP-net. We first process the task parameter input $\theta$, an image encoding the environment and goal, by a CNN. We separately process the action $\underline{a}_t$ and observation $\underline{o}_t$ input vectors by a two-layer fully connected component. These processed inputs are concatenated into a single vector which is the input of the LSTM layer. The size of the LSTM hidden state and output is chosen to match the number of states in the grid, e.g. $N^2$ for an $N \times N$ grid. We initialize the hidden state of the LSTM using the appropriate channel of the input $\theta$ that encodes the initial belief.

**CNN+LSTM.** CNN+LSTM is a state-of-the-art deep convolutional network with LSTM cells. It is similar in structure to DRQN [10], which was used for learning to play partially observable Atari games in a reinforcement learning setting. Note that we train the networks in an imitation learning setting using the same set of expert trajectories, and not using reinforcement learning, so the comparison with QMDP-net is fair. The CNN+LSTM network has more structure to encode a decision making policy compared to a vanilla RNN, and it is also more tailored to our input representation. We process the image input, $\boldsymbol{\theta}$, by a CNN component and the vector input, $\underline{a}_t$ and $\underline{o}_t$, by a fully connected network component. The output of the CNN and the fully connected component are then combined into a single vector and fed to the LSTM layer.

**RNN.** The considered RNN architecture is a vanilla recurrent neural network with $512$ hidden units and tanh activation. At each step inputs are transformed into a single concatenated vector. The outputs are obtained by a fully connected layer with softmax activation.

We performed hyperparameter search on the number of layers and hidden units, and adjusted learning rate and batch size for all alternative networks. In particular, we ran trials for the deterministic grid navigation task. For each architecture we chose the best parametrization found. We then used the same parametrization for all tasks.

## C.7 Training Technique

We train all networks, QMDP-net and alternatives, in an imitation learning setting. The loss is defined as the cross-entropy between predicted and demonstrated actions along the expert trajectories. We do not receive supervision on the underlying ground-truth POMDP models.

We train the networks with backpropagation through time on mini-batches of $100$. The networks are implemented in Tensorflow [1]. We use RMSProp optimizer [35] with $0.9$ decay rate and $0$ momentum setting. The learning rate was set to $1 \times 10^{-3}$ for QMDP-net and $1 \times 10^{-4}$ for the alternative networks. We limit the number of backpropagation steps to $4$ for QMDP-net and its untied variant; and to $6$ for the other alternatives, which gave slightly better results. We used a combination of early stopping with patience and exponential learning rate decay of $0.9$. In particular, we started to decrease the learning rate if the prediction error did not decrease for $30$ consecutive epochs on a validation set, $10\%$ of the training data. We performed $15$ iterations of learning rate decay.

We perform multiple rounds of the training method described above. In our partially observable domains predictions are increasingly difficult along a trajectory, as they require multiple steps of filtering, i.e. integrating information from a long sequence of observations. Therefore, for the first round of training we limit the number of steps along the expert trajectories, for training both QMDP-net and its alternatives. After convergence we perform a second round of training on the full length trajectories. Let $L_r$ be the number of steps along the expert trajectories for training round $r$. We used two training rounds with $L_1 = 4$ and $L_2 = 100$ for training QMDP-net and its untied variant. For training the other alternative networks we used $L_1 = 6$ and $L_2 = 100$, which gave better results.

We trained policies for the grid navigation task when the grid is fixed, only the initial state and goal vary. In this variant we found that a low $L_r$ setting degrades the final performance for the alternative networks. We used a single training round with $L_1 = 100$ for this task.