[Reviews · NeurIPS 2017]

Reviewer 1



The paper introduces a policy network architecture, the QMDP-net, that explicitly contains a separate filter and planning module, where the filter is needed for environments with partial observability. The QMDP-net is tested on multiple partially observable problems, where it has good performance on all of them. The paper is an interesting addition to the literature on training policy networks, where the network structure is designed in a way to introduce a "prior" on the type of computations the network is expected to need to fulfil its task. The partially observable markov decision process is a problem type that has a clear significance also in practical cases. The authors make a good effort at comparing their model to alternative architectures, such as a generic RNN and CNN+LSTM. I think the paper could have been even stronger if the authors had found some problem that would already have been tackled by others, and compare their model to that. I am not familiar enough with the literature to know whether a good such set exists, though. In general the paper is well written and seems methodologically sound. As a minor comment, the abstract seems to be in a smaller font (possibly due to the italicised font) than the rest of the text, I would consider revising that.

Reviewer 2



The paper proposes a novel policy network architecture for partially observable environments. The network includes a filtering module and a planning module. The filtering module mimics computation of the current belief of the agent given its previous belief, the last action and the last observation. The model of the environment and the observation function are replaced with trainable neural modules. The planning module runs value iteration for an MDP, whose transition function and reward function are also trainable neural modules. The filtering model and the planning module are stacked to resemble the QMDP algorithm for solving POMDP. All the modules are trained jointly according to an imitation learning criterion, i.e. the whole network is trained to predict the next action of an expert given the past observations and actions. The experimental evaluation shows better generalization to new tasks. The approach can be as an extension of Value Iteration Networks (VIN) to partially observable environments by means of adding the filtering network. The paper is in general well written. I did not understand though why the internal model of the network does not use the same observation and action spaces as the real model of the world. Sentences like "While the belief image is defined in M, action inputs are defined in \hat{M}" were therefore very hard to understand. The choice of notation f^t_T is quite confusing. The experimental evaluation seems correct. I would suggest a convolutional LSTM as another baseline for the grid world. A fully-connected LSTM is obviously worse eqipped than the proposed model to handle their local structure. I am somewhat suspicious if the fact that expert trajectories came from QMDP could give QMDP-net an unfair advantage, would be good if the paper clearly argued why it is not the case. One limitation is this line of research (including VIN) is that it assumes discrete state spaces and the amount of computation is proportional to the number of states. As far as I can I see this makes this approach not applicable to real-world problems with high-dimensional state spaces. Overall, the paper proposes a novel and elegant approach to build policy networks and validates it experimentaly. I recommend to accept the paper.

Reviewer 3



This paper proposes a deep learning system for planning in POMDPs. The system attempts to combine model-based and model-free methods in an end-to-end differentiable recurrent policy network. The paper suffers from circuitous writing. Inexplicably, the authors are not upfront about the problem addressed. For example, they don't reveal until the bottom of the third page that this work primarily addresses imitation learning and not the general planning setting. Notation is also introduced a bit clumsily. The authors devise an architecture based on the QMDP algorithm. The architecture consists of one component which is meant to calculate an agents belief over the current state at any step and another that given these beliefs generates actions. The ideas are interesting but they could be presented much more clearly. I would have much preffered for the authors to just enumerate what precisely takes place at each step of training. A model with this many non-standard components requires clearer explanation. I suspect the authors are doing some interesting work here but cannot accept the paper in its current form. I suggest that the authors give this paper a full rewrite and articulate precisely what they do, sufficiently clearly that any expert reader could re-implement the work from scratch.